# Amplify Adjacent Token Differences: Enhancing Long Chain-of-Thought Reasoning with Shift-FFN

## Abstract

Recently, models such as OpenAI-o1 and DeepSeek-R1 have demonstrated remarkable performance on complex reasoning tasks through Long Chain-of-Thought (Long-CoT) reasoning. Although distilling this capability into student models significantly enhances their performance, this paper finds that fine-tuning LLMs with full parameters or LoRA with a low rank on long CoT data often leads to *Cyclical Reasoning*, where models repeatedly reiterate previous inference steps until the maximum length limit. Further analysis reveals that smaller differences in representations between adjacent tokens correlates with a higher tendency toward *Cyclical Reasoning*. To mitigate this issue, this paper proposes Shift Feedforward Networks (Shift-FFN), a novel approach that edits the current token's representation with the previous one before inputting it to FFN. This architecture dynamically amplifies the representation differences between adjacent tokens. Extensive experiments on multiple mathematical reasoning tasks demonstrate that LoRA combined with Shift-FFN achieves higher accuracy and a lower rate of *Cyclical Reasoning* across various data sizes compared to full fine-tuning and standard LoRA. Our data and code are available at `https://anonymous.4open.science/r/Shift-FFN`.

## 1 Introduction

In recent years, Large Reasoning Models (LRMs) such as OpenAI-o1 Jaech et al. (2024), DeepSeek-R1DeepSeek-AI et al. (2025), and Qwen3 Yang et al. (2024) have achieved significant advancements in mathematical, coding, and other complex reasoning tasks. A key factor behind their success lies in extending the traditional Chain-of-Thought (CoT) Wei et al. (2023) approach into long CoT, which incorporates detailed step-by-step reasoning, multiple solution strategies and self-reflection processes Chen et al. (2025).

Long Chain-of-Thought (CoT) demonstrates remarkable reasoning abilities, but training language models to exhibit such complex reasoning requires substantial computational resources. Consequently, knowledge distillation (Xu et al., 2024b; Ye et al., 2025; Li et al., 2025; Luo et al., 2025) has emerged as a prevalent approach to impart this extended reasoning capabilities to smaller models by training them on instruction-response instances enriched with short/long CoT (short/long CoT datasets for short). Therefore, how to enable student models to effectively learn from these long CoT datasets has emerged as a fundamental scientific problem.

One potential method could be Parameter-Efficient Fine-Tuning (PEFT) (Han et al., 2024) such as LoRA Hu et al. (2021), which achieves performance comparable to full fine-tuning on several tasks such as commonsense reasoning Clark et al. (2019), and instruction following Li et al. (2023), despite updating only a minimal subset of parameters. However, these tasks typically do not involve long CoT reasoning and generally maintain output lengths within only 1k tokens, whereas long CoT data frequently exceed 16k tokens. This discrepancy naturally raises a critical research question: **Is the PEFT approach still effective when applied to learning long CoT reasoning?**

This paper first investigates this question by constructing parallel datasets containing short CoT and long CoT solutions for identical questions, respectively. The short and long CoT datasets are sourced from Numina Math dataset LI et al. (2024) and DeepSeek-R1 outputs, respectively. Two student models are trained separately using LoRA Hu et al. (2021) and full fine-tuning, respectively. This

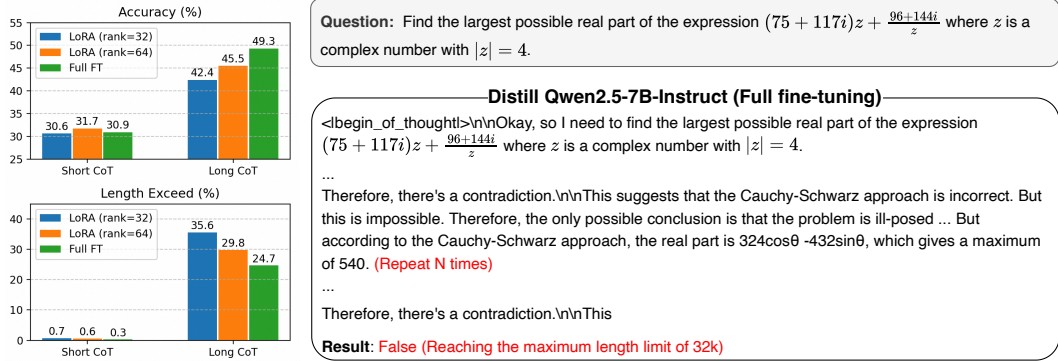

Figure 1: (**Left**), performance comparison of LoRA and Full Fine-Tuning (Full FT) on Accuracy (%) and Length Exceed (%) metrics for short CoT and long CoT datasets. "Accuracy" represents the average accuracy across four mathematical tasks. "Length Exceed" indicates the percentage of model outputs that exceed the maximum length limit. (**Right**), an example of *Cyclical Reasoning*.

paper observes that with a rank of 32, LoRA achieves comparable performance to full fine-tuning on short CoT dataset. However, a noticeable performance gap emerges between LoRA and full fine-tuning in long CoT scenarios, as shown in Figure 1 (left). This paper finds that both LoRA and full fine-tuned models tend to exhibit *Cyclical Reasoning*, where they repeatedly generate paragraphs or reiterate previous inference steps until reaching the maximum length limit of 32k tokens, Figure 1 (right). This phenomenon is more pronounced in LoRA with lower rank, contributing to the performance gap compared to full fine-tuning. Further analysis reveals that low divergence of adjacent tokens correlates with a higher tendency toward *Cyclical Reasoning*. Specifically, this paper finds that: (1) For the same model, answers exhibiting *Cyclical Reasoning* show smaller internal representation differences between adjacent tokens compared to normal answers. (2) For LoRA fine-tuned models, a higher rank reduces the rate of *Cyclical Reasoning* while simultaneously increases the internal representation differences between adjacent tokens (more details in Section 3.1).

Based on these observations, the paper proposes Shift Feedforward Network (Shift-FFN), which introduces an Editor module before the FFN. The Editor module uses the preceding token's representation to edit the current token's representation, thereby dynamically amplifying the representation differences between adjacent tokens within the model, as shown in Figure 3. Experimental results demonstrate that LoRA combined with Shift-FFN achieves higher accuracy and a lower rate of *Cyclical Reasoning* across various data sizes compared to full fine-tuning and standard LoRA.

The main contributions of this work are as follows:

1. This paper finds that fine-tuning LLMs with full parameters or LoRA with a low rank on long CoT data often leads to *Cyclical Reasoning*, and observes smaller differences in representations between adjacent tokens correlates with a higher tendency toward *Cyclical Reasoning*.

2. This paper proposes Shift-FFN, which edits the current token's representation with the previous one before FFN, thereby dynamically amplifying differences between adjacent tokens.

3. Experimental results show that introducing Shift-FFN into LoRA improves model accuracy and reduces the ratio of *Cyclical Reasoning*.

## 2 RELATED WORK

**Parameter-Efficient Fine-Tuning methods (PEFTs).** PEFT methods adapt models to downstream tasks by updating only a small subset of parameters. Existing PEFT methods can be categorized into the following three categories Han et al. (2024):

1. **Addition based methods** train additional lightweight modules that are positioned within the frozen model. Adapters insert small adapter layers between LM attention or MLP layers

(Houlsby et al., 2019; Wang et al., 2022; He et al., 2022). Prompt tuning inserts randomly-initialized soft tokens at the beginning of the input texts and trains their embeddings while keeping the LM weights frozen (Lester et al., 2021; Li & Liang, 2021).

2. **LoRA** Hu et al. (2021) and its variants Zhang et al. (2023); Liu et al. (2023b) employ low-rank matrix approximations for weight updates during training, while introducing no inference overhead as the updates can be directly merged into the base model parameters.

3. **Representation editing based methods** are motivated by representation engineering which demonstrates that adding "steering vectors" to the representation of each hidden layer can control pretrained LM generations Subramani et al. (2022); Liu et al. (2023a); Tang et al. (2025). Therefore, these methods learn to modify the hidden representations generated by multi-head attentions or FFNs (Liu et al., 2023c; Wu et al., 2024a;b)

Our proposed Shift-FFN can be viewed as a representation editing-based method, but it incorporates preceding token information in the updating of representation.

**Long CoT Distillation**. Extensive studies have demonstrated that distilling long CoT data from powerful reasoning models into student models can significantly enhance the students' reasoning capabilities DeepSeek-AI et al. (2025); Yang et al. (2024); Wen et al. (2025). Furthermore, LIMO Ye et al. (2025) reveals that a small set of carefully selected examples suffices to elicit the model's complex mathematical reasoning capabilities. Li et al. (2025) finds that the structure of long CoT proves essential for effective learning, while the specific content within individual reasoning steps exhibits minimal impact. DLCoT Luo et al. (2025) proposes to optimize long CoT through segmentation, redundancy elimination, and error correction. Their experimental results demonstrated that eliminating redundant reasoning paths leads to improvements in distillation efficiency. While existing approaches primarily investigate from a data perspective, this paper focuses on model architecture, enabling Shift-FFN to be complementary with such methods.

**Token Shift**. RWKV Peng et al. (2023) introduces time-mixing and channel-mixing by computing linear projections from weighted combinations of the current and previous input representations within each block. KV shift Xu et al. (2024a) performs linear combinations of the current token's key/value vectors with those of the preceding token, and demonstrates that Shift-KV attention exhibits enhanced capability in learning induction heads. Fox Lin et al. (2025) dynamically computes the weighting coefficient for the preceding token in the shift operation, followed by RMSNorm (Root Mean Square Normalization) Zhang & Sennrich (2019) of the weighted results. These methods focus on training a model from scratch, whereas this paper studies how to fine-tune a model better by shifting tokens.

## 3 METHOD

### 3.1 MOTIVATION

**Feature Definition.** Wang et al. (2025) explores the internal workings of LLMs by treating the sequence of hidden states as a Chain-of-Embedding (CoE), representing the model's latent thought process. Their analysis reveals distinct patterns in these CoE features when LLMs produced correct versus incorrect answers. Motivated by this work, we pose the question: Can the internal hidden states of a model be leveraged to detect instances of *Cyclical Reasoning*?

Instead of averaging token representations per layer and forming an embedding trajectory from these layer-wise averages (Wang et al., 2025; 2024), we utilize the sequence of token representations from each layer as our embedding trajectory. The embedding trajectory at layer $l$, denoted as $\boldsymbol{X}^l$, is formalized as follows:

$$\boldsymbol{X}^l = \boldsymbol{x}_0^l \to \boldsymbol{x}_1^l \to ... \to \boldsymbol{x}_{I-1}^l \to \boldsymbol{x}_I^l \tag{1}$$

where $\boldsymbol{x}_i^l$ denotes the hidden state of the $i$-th token after attention in the $l$-th layer, $I$ is the length of the generated sequence. We measure the LLMs' thinking process by using the relative change in hidden states at each time step.

$$s(\boldsymbol{x}_{i-1}^l, \boldsymbol{x}_i^l) = \frac{\|\boldsymbol{x}_i^l - \boldsymbol{x}_{i-1}^l\|_2}{\|\boldsymbol{x}_{i-1}^l\|_2} \tag{2}$$

Then we define the overall relative change of the embedding trajectories, denoted as $M(\boldsymbol{X})$, as the average of the relative changes between every adjacent tokens across all layers. This can be formalized as follows:

$$M(\boldsymbol{X}) = \frac{1}{L \times I} \sum_{l=1}^{L} \sum_{i=1}^{I} s(\boldsymbol{x}_{i-1}^l, \boldsymbol{x}_i^l) \tag{3}$$

where $L$ is the total number of layers in the LLM, $I$ is the length of the generated sequence.

**Analysis Setup and Findings.** We train two models on a 20k long CoT using LoRA and full fine-tuning, respectively. We evaluate these models on a randomly selected set of 100 questions from the MATH500 Hendrycks et al. (2021), with a sampling of eight times, and exclude questions where all eight generated responses exceed the maximum length limit. For the remaining length-exceeded responses, we truncate them to the average length of the normal (non-length-exceeded) responses and remove all repeated text segments. Finally, we calculate the $M(\boldsymbol{X})$ values for both the normal and the length-exceeded responses. The results are shown in Figure 2, we can find that the "Exceed" samples tend to exhibit a lower mean relative change compared to the "Normal" samples in

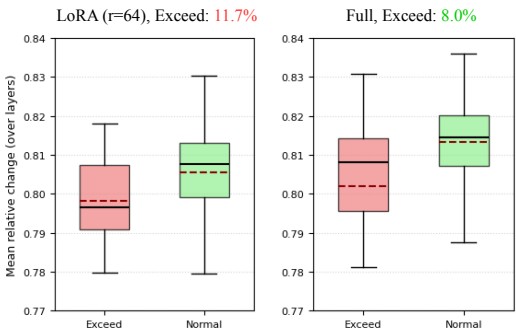

Figure 2: Distribution of the $M(\boldsymbol{X})$ for Exceed and Normal samples, comparing LoRA and Full fine-tuned models. The dashed red line represents the mean value.

both models, as indicated by the lower median and mean (dashed red line) of the "Exceed" box plots. This suggests that when the models engage in *Cyclical Reasoning* (section 4.2 elaborates on the rationale for using the *Length Exceeded Percentage* to measure *Cyclical Reasoning*), the relative change in their adjacent hidden states tends to be less pronounced on average. Furthermore, this paper finds that the full fine-tuned model exhibits a lower proportion of Exceed samples, and concurrently, the $M(\boldsymbol{X})$ value across all its generated samples is also higher.

Based on these findings, a natural research question arises: **Can we mitigate models' *Cyclical Reasoning* issues and consequently enhance its performance by dynamically amplifying representation differences between adjacent tokens?**

## 3.2 SHIFT FEEDFORWARD NETWORK

Motivated by the aforementioned considerations, we propose Shift Feedforward Network (Shift-FFN), an architecture that introduces an Editor module before the FFN. This module uses the preceding token's representation to edit the current token's representation, thereby dynamically amplifying the representation differences between adjacent tokens. The mathematical formulation of this process is as follows:

$$\text{Shift-FFN}(\boldsymbol{x_i}) = \text{FFN}(\boldsymbol{x_i} + f_s(\boldsymbol{x_{i-1}}, \boldsymbol{x_i})) \tag{4}$$

where FFN is the original feedforward layer, $f_x(\cdot)$ represents shift function, which is defined as:

$$f_s(\boldsymbol{x_{i-1}}, \boldsymbol{x_i}) = W_c \left[ ReLU(W_b \left[ \boldsymbol{x_{i-1}}; \boldsymbol{x_i} \right]) \odot (W_a \, \boldsymbol{x_{i-1}}) \right] \tag{5}$$

where $\boldsymbol{x_i} \in \mathbb{R}^d$ is the representation of token $i$ after attention, $[\,;]$ denotes concatenate operation, $W_b \in \mathbb{R}^{r \times 2d}$, $W_a \in \mathbb{R}^{r \times d}$ and $W_c \in \mathbb{R}^{d \times r}$ are parameter matrices in the Editor module, and they are trained from scratch. To maintain a manageable increase in the number of parameters, we set the dimensionality $r$ to be significantly smaller than $d$ ($r \ll d$), In LoRA fine-tuning, the value of $r$ corresponds to the rank of the LoRA. To ensure training stability in the initial stages, we initialize the matrix $W_c$ as an all-zero matrix. This initialization causes the Shift-FFN to degenerate into the original FFN during the early phase of training.

## 3.3 ANALYSIS

From simplicity, we use the standard $\text{FFN}(x_i) = W_{down}[\sigma(W_{up} \, x_i)]$ and simplify $f_s(\boldsymbol{x_{i-1}}, \boldsymbol{x_i})$ as follows.

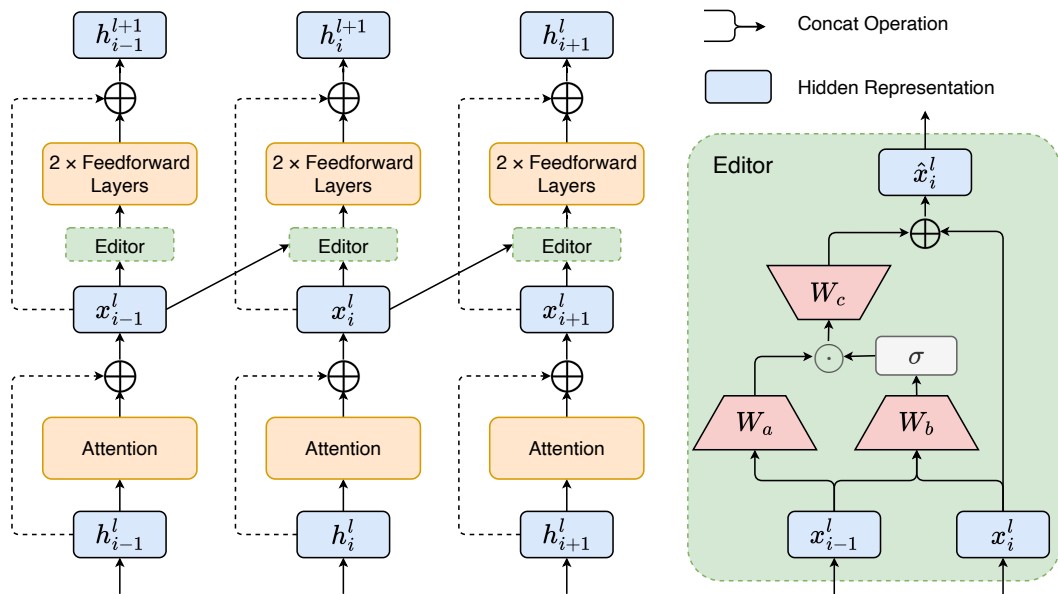

Figure 3: The architecture of Shift-FFN, the **left** side describes the process of shifting token, the **right** side demonstrates the detail of the Editor module. $\sigma$ is the ReLU function. $\odot$ and $\oplus$ are element-wise multiplication and addition, respectively.

$$f_s(\boldsymbol{x_{i-1}}, \boldsymbol{x_i}) = W_c W_b \boldsymbol{x}_{i-1} = W_s \boldsymbol{x}_{i-1} = \hat{\boldsymbol{x}}_{i-1} \tag{6}$$

**From the Perspective of Self Attention**. As defined previously, the output of the Shift-FFN can be expressed as:

$$\boldsymbol{h}_i^{l+1} = W_{down}[\sigma(W_{up}(\boldsymbol{x}_i + \hat{\boldsymbol{x}}_{i-1})] = \boldsymbol{h}_i + \hat{\boldsymbol{h}}_i \tag{7}$$

where $\boldsymbol{h}_i$ is the original FFN output, $\hat{\boldsymbol{h}}_i = W_{down}[\sigma(W_{up}\hat{\boldsymbol{x}}_{i-1})]$ is introduced by Shift-FFN additionally. Then, the attention score $\alpha_{i,j}$ between token $i$ and $j$ at layer $l + 1$ is calculated as follows (residual connections and normalization are omitted):

$$\alpha_{i,j} = [W_q(\boldsymbol{h}_i + \hat{\boldsymbol{h}}_i)]^T [W_k(\boldsymbol{h}_j + \hat{\boldsymbol{h}}_j)] \tag{8}$$
$$= \alpha'_{i,j} + (W_q\boldsymbol{h}_i)^T(W_k\hat{\boldsymbol{h}}_j) + (W_q\hat{\boldsymbol{h}}_i)^T(W_k\boldsymbol{h}_j) + (W_q\hat{\boldsymbol{h}}_i)^T(W_k\hat{\boldsymbol{h}}_j)$$

where $W_q$ and $W_k$ denote the Query and Key parameter matrices at layer $l + 1$, $\alpha'_{i,j} = (W_q\boldsymbol{h}_i)^T(W_k\boldsymbol{h}_j)$ is the original attention score, and we have

$$(W_q\boldsymbol{h}_i)^T(W_k\hat{\boldsymbol{h}}_j) = \boldsymbol{h}_i^T W_q^T W_k W_{down}[\sigma(W_{up}\hat{\boldsymbol{x}}_{j-1})] \tag{9}$$

Let $A_i = \boldsymbol{h}_i^T W_q^T W_k W_{down}$. Finally, neglecting the higher-order infinitesimal terms, and substituting $\hat{\boldsymbol{x}}_{i-1} = W_s \boldsymbol{x}_{i-1}$, we can express $\alpha_{ij}$ as:

$$\alpha_{i,j} = \alpha'_{i,j} + A_i[\sigma(W_{up} W_s \boldsymbol{x}_{j-1})] + A_j[\sigma(W_{up} W_s \boldsymbol{x}_{i-1})] \tag{10}$$

As evident from the derived formulas, the Shift-FFN effectively augments the original attention score with a correction term that is contingent on the $(i - 1)$-th and $(j - 1)$-th tokens.

## 4 EXPERIMENT

### 4.1 EXPERIMENT SETUP

**Training Data.** To compare the models' performance under short CoT and long CoT conditions, we specifically select the mathematics portion of the OpenThoughts dataset Team (2025), which

Table 1: Performance of models on mathematical reasoning benchmarks with different training setups. Each cell presents the *Accuracy* followed by the *Length Exceeded Percentage $P_E$* (in parentheses) which indicates the percentage of generated responses exceeding the 32k token limit. The "Param" column indicates the number of trainable parameters. The best performance within each LoRA configuration is highlighted in bold.

| | Method | Param | AIME24 | AMC23 | MATH500 | Olympiad | Avg |
|---|---|---|---|---|---|---|---|
| | Full | 3.09B (100%) | 3.1 (80.3) | 29.4 (53.2) | 51.2 (34.8) | 20.2 (55.5) | 26.0 (55.9) |
| | LoRA (r=128) | 0.24B (7.8%) | 4.3 (62.3) | 30.6 (40.1) | 54.4 (25.1) | 21.9 (41.7) | 27.8 (42.3) |
| Qwen2.5-3B | LoRA+Shift-FFN (r=128) | 0.28B (9.1%) | **4.6** (57.1) | **31.5** (34.7) | **55.0** (21.2) | **23.7** (37.0) | **28.7** (37.5) |
| | LoRA (r=256) | 0.48B (15.6%) | 5.4 (49.4) | 32.7 (31.6) | 57.6 (17.2) | 24.1 (31.3) | 30.0 (32.3) |
| | LoRA+Shift-FFN (r=256) | 0.55B (18.2%) | **7.0** (43.2) | **35.2** (24.9) | **60.2** (13.9) | **25.6** (28.2) | **32.0** (27.5) |
| | Full | 8.03B (100%) | 6.7 (23.3) | 41.4 (13.5) | 63.2 (5.3) | 30.7 (12.6) | 35.5 (13.7) |
| | LoRA (r=128) | 0.34B (4.2%) | **4.6** (35.7) | 34.0 (22.1) | 58.2 (9.3) | 26.0 (22.2) | 30.7 (22.4) |
| Llama3.1-8B | LoRA+Shift-FFN (r=128) | 0.40B (5.0%) | 3.6 (34.5) | **34.3** (17.6) | **60.2** (9.0) | **27.0** (18.3) | **31.3** (19.8) |
| | LoRA (r=256) | 0.67B (8.4%) | **5.4** (25.9) | 37.8 (15.1) | 62.5 (6.6) | 29.3 (15.7) | 33.7 (15.8) |
| | LoRA+Shift-FFN (r=256) | 0.81B (10.0%) | 5.1 (22.8) | **38.0** (12.1) | **63.2** (5.1) | **29.4** (13.7) | **34.0** (13.4) |
| | Full | 7.62B (100%) | 20.0 (42.3) | 58.1 (17.3) | 78.7 (8.0) | 42.1 (23.6) | 49.3 (24.7) |
| | LoRA (r=128) | 0.32B (4.2%) | 17.8 (42.5) | 54.7 (20.2) | 76.1 (8.2) | 39.9 (24.1) | 47.1 (23.7) |
| Qwen2.5-7B | LoRA+Shift-FFN (r=128) | 0.37B (4.9%) | **18.2** (35.6) | **55.6** (15.3) | **78.1** (7.0) | **41.0** (19.1) | **48.2** (19.2) |
| | LoRA (r=256) | 0.64B (8.4%) | 21.0 (28.6) | 58.5 (10.9) | 79.1 (5.2) | 43.0 (15.2) | 50.4 (15.0) |
| | LoRA+Shift-FFN (r=256) | 0.75B (9.8%) | **21.8** (23.5) | **59.1** (9.9) | **79.9** (4.1) | **43.8** (13.1) | **51.2** (12.7) |

collects long CoT from DeepSeek-R1 DeepSeek-AI et al. (2025). Our short CoT data is from the Numina-Math dataset LI et al. (2024). Additionally, we exclude OpenThoughts samples with response lengths exceeding 16k to prevent our models from learning incomplete reasoning processes. After this filtering, we retain a total of 89k training examples, from which we randomly sample 20k for our main experiment.

**Training Setup.** We utilize the LlamaFactory framework Zheng et al. (2024) and LoRA Hu et al. (2021) to fine-tune the Qwen2.5-3B-Instruct, Qwen2.5-7B-Instruct and Llama3.2-8B-Instruct with a batch size of 96 and a learning rate of 1e-4, employing a warm-up ratio of 0.1 and a linear learning rate decay schedule, similar to Li et al. (2025). For full fine-tuning, we maintain the same hyperparameters except for a learning rate of 1e-5. The max sequence length is set to 16k for all training. All experiments are conducted on $8 \times 80G$ L20 GPUs.

**Evaluation Setup.** We evaluate our models on four mathematical reasoning datasets: AIME24, AMC23, MATH500 Hendrycks et al. (2021), and OlympiadBench He et al. (2024). We use a sampling temperature of 0.6 and set the maximum generation length to 32k tokens. To mitigate the impact of randomness in the results, we average over 32 runs for AIME and AMC, and 4 runs for the other tasks.

## 4.2 MAIN RESULTS

Table 1 presents the results of full fine-tuning and LoRA fine-tuning (with and without Shift-FFN) for various models. The results reveal several findings as follows:

**Long CoT Learning Requires Higher LoRA Rank**. We find that in long CoT scenarios, achieving performance with LoRA comparable to full fine-tuning necessitates a higher LoRA rank, such as 256, in contrast to simpler tasks like common-sense reasoning where a much lower rank (e.g., 32) often suffices to approximate full fine-tuning performance.

**Cyclical Reasoning**. We quantify the *Cyclical Reasoning* by using the *Length Exceeded Percentage* (denoted as $P_E$) – the proportion of generated samples exceeding the 32k token limit. Given the maximum training sequence length of 16k, a 32k limit during inference is ample for generating correct answers; therefore, exceeding this limit is considered indicative of the model getting stuck in a loop. We further analyze the proportion of repetitive output within these length-exceeded samples, where the model repeatedly generates the same segment of text until the maximum limit

Table 2: Comparison of models' performance w.t./w.o. Shift-FFN under comparable trainable parameters. Each cell presents the *Accuracy* followed by the *Length Exceeded Percentage $P_E$* (in parentheses) which indicates the percentage of generated responses exceeding the 32k token limit.

| Method | Param | AIME24 | AMC23 | MATH500 | Olympiad | Avg |
|---|---|---|---|---|---|---|
| LoRA (r=128) | 0.32B (4.2%) | 17.8 (42.5) | 54.7 (20.2) | 76.1 (8.2) | 39.9 (24.1) | 47.1 (23.7) |
| LoRA (r=148) | 0.37B (4.9%) | 17.5 (40.9) | 55.3 (18.2) | 76.8 (8.6) | 40.6 (22.1) | 47.5 (22.4) |
| LoRA+Shift-FFN (r=128) | 0.37B (4.9%) | **18.2** (35.6) | **55.6** (15.3) | **78.1** (7.0) | **41.0** (19.1) | **48.2** (19.2) |
| LoRA (r=256) | 0.64B (8.4%) | 21.0 (28.6) | 58.5 (10.9) | 79.1 (5.2) | 43.0 (15.2) | 50.4 (15.0) |
| LoRA (r=296) | 0.75B (9.8%) | 21.2 (28.4) | 58.5 (13.6) | 79.3 (6.0) | 43.2 (15.5) | 50.6 (15.9) |
| LoRA+Shift-FFN (r=256) | 0.75B (9.8%) | **21.8** (23.5) | **59.1** (9.9) | **79.9** (4.1) | **43.8** (13.1) | **51.2** (12.7) |

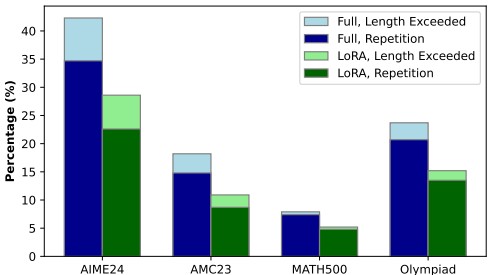

Figure 4: Proportion of length-exceeded and repetition samples in different models.

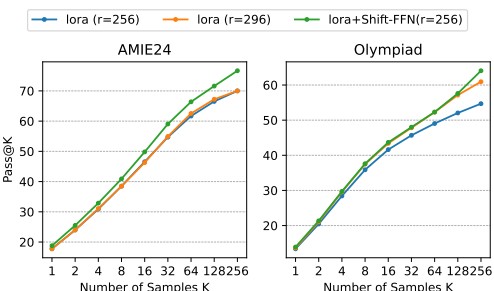

Figure 5: Pass@K of models with different training setups on AIME24 and OlympiadBench.

is reached. The results of this analysis are presented in Figure 4. We find that over 80% of the length-exceeded samples exhibit exact textual repetition. While the remaining 20% do not show identical text repetition, they still demonstrate patterns of *Cyclical Reasoning*, such as repeatedly verifying the same step or iterating through the same few inference steps, concrete examples can be found in Appendix C. Therefore, utilizing the $P_E$ as a metric for *Cyclical Reasoning* is a justifiable approach. Using this metric, we find that models trained on long CoT data tend to exhibit *Cyclical Reasoning*. Even the full fine-tuned Qwen2.5-7B-Instruct shows a 24.7% *Cyclical Reasoning* ratio. When using LoRA fine-tuning, this ratio decreases as the rank increases. Interestingly, we find that LoRA fine-tuned Qwen2.5-7B-Instruct with a rank of 256 significantly reduces the *Cyclical Reasoning* ratio by 12% compared to full fine-tuning.

**Effectiveness of Shift-FFN**. It can be found that the integration of Shift-FFN consistently yields performance improvements across all settings. Specifically, the Qwen2.5-7B-Instruct model trained with LoRA at rank 256 already achieves an average accuracy 0.9% higher than the full fine-tuned model. Upon introducing Shift-FFN, the model's average performance further improves by 0.8% to 51.2%, surpassing the full fine-tuned baseline and the original LoRA model across all datasets. Furthermore, Shift-FFN not only enhances performance but also significantly reduces *Cyclical Reasoning*, which is reflected by the decreasing of $P_E$ from 15.0% to 12.7%.

### 4.3 COMPARED TO LoRA WITH THE SAME NUMBER OF PARAMETERS

As Shift-FFN introduces extra parameters, to compare it more fairly with standard LoRA, we increase LoRA's rank (e.g., from 256 to 296) in the training of Qwen2.5-7B-Instruct. This makes the total number of parameters the same as LoRA+Shift-FFN. Table 2 shows the experimental results. It can be found that compared to simply increasing the rank, introducing Shift-FFN brings a larger improvement with a similar number of added parameters. Specifically, when the rank is 256, increasing it to 296 only slightly improves the average performance from 50.4% to 50.6% and also increases the $P_E$. However, introducing Shift-FFN raises it to 51.2% and also further reduces the $P_E$. A possible explanation is that at a rank of 256, LoRA is nearing its performance limit, so further increasing

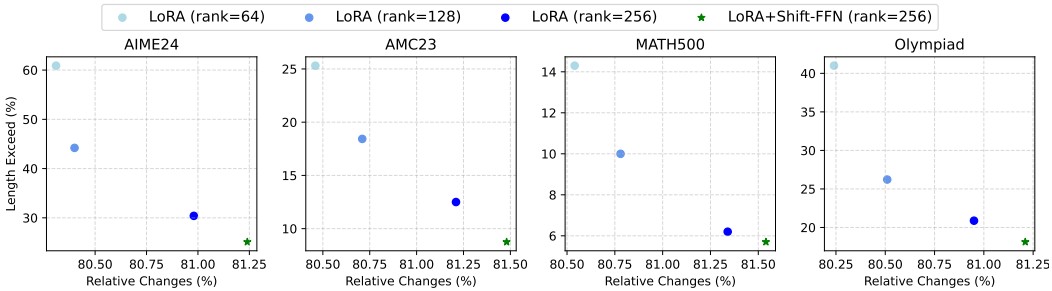

Figure 6: Comparison of *Mean Relative Change* $M(\boldsymbol{X})$ and *Length Exceeded Percentage* $P_E$ for non-length-exceeded samples across models trained with different settings on four datasets.

the rank yields diminishing returns. **However, introducing Shift-FFN can further improve the model's performance limit from the perspective of representation learning.**

To further validate the effectiveness of Shift-FFN, we evaluate the pass@K metric on the AMIE 24 and OlympiadBench datasets. For computational efficiency, we select the first 64 questions from OlympiadBench and set the maximum generation length to 16k. The results of these experiments are presented in Figure 5. It demonstrates that incorporating Shift-FFN leads to improvements across all pass@K metrics. Specifically, on the AIME 24 dataset, pass@256 increases from 70.0% to 76.7% with the addition of Shift-FFN. A potential reason for this is that Shift-FFN reduces the tendency of the model to engage in *Cyclical Reasoning* ($P_E$ decreases from 28.4% to 23.5%), thereby enhancing the model's exploration efficiency. On OlympiadBench, the $P_E$ only decreases by 2.1% with the integration of Shift-FFN. Consequently, the difference in pass@K is not significant for $K \leq 64$. The performance gap only becomes more apparent as K increases further. Shift-FFN also consistently achieves the best performance across different sampling temperatures, more details can be found in Appendix B.

### 4.4 MEAN RELATIVE CHANGES WITH SHIFT-FFN

To further investigate the relationship between *Mean Relative Change* $M(\boldsymbol{X})$ and *Length Exceeded Percentage* $P_E$, as well as the impact of Shift-FFN, we analyze these metrics for Qwen2.5-7B-Instruct with different training settings across the datasets, as shown in Figure 6. We find that as the LoRA rank increases, the model's $M(\boldsymbol{X})$ also increases, while $P_E$ decreases correspondingly. This indicates a negative correlation between $M(\boldsymbol{X})$ and $P_E$. Specifically, for the AIME24 dataset, when the rank increases from 64 (light blue point) to 256 (dark blue point), $M(\boldsymbol{X})$ increases from 80.31% to 80.98%, and $P_E$ correspondingly decreases from 60.9% to 30.4%. This suggests that as the model has more trainable parameters in the LoRA settings, it becomes less prone to generating *Cyclical Reasoning*, and the relative changes between its internal adjacent tokens become more pronounced. The introduction of Shift-FFN consistently achieves the lowest $P_E$ and the highest $M(\boldsymbol{X})$. For example, on the AIME24 dataset, introducing Shift-FFN increases $M(\boldsymbol{X})$ from 80.98% to 81.24%,and also further reduces $P_E$ from 30.4% to 25.1%. Furthermore, we find that the higher the original $P_E$ of the model on a dataset, the greater the benefit brought by introducing Shift-FFN. This indicates that Shift-FFN effectively mitigates the issue of *Cyclical Reasoning* by enabling a dynamic amplifying the representation differences between adjacent tokens.

### 4.5 PERFORMANCE OF SHIFT-FFN WITH VARYING TRAINING DATA SIZES

To evaluate the performance of Shift-FFN with varying training data sizes, we randomly sample 10k, 20k, 40k, and 80k examples from OpenThoughts for training. For each data size, we train three models: Full fine-tuning, LoRA (r=296), and LoRA+Shift-FFN (r=256). The results are depicted in Figure 7. We notice that as the training sample size increases, the performance of all models improves, and the $P_E$ decreases. Interestingly, LoRA fine-tuned models consistently outperform the full fine-tuned model across all data scales and are less prone to generating length-exceeded outputs, particularly with smaller training datasets. Specifically, with only 10k training samples, the full fine-tuned model shows a 38.0% of $P_E$, while the LoRA fine-tuned model exhibits only

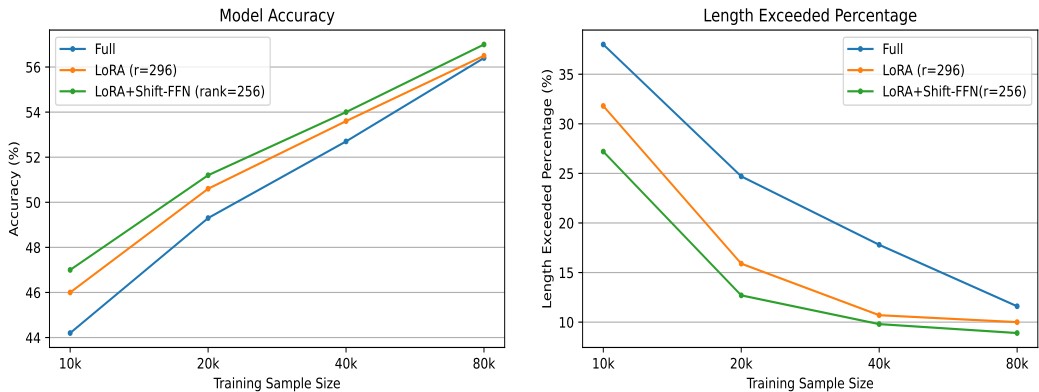

Figure 7: The *Accuracy* (left) and the *Length Exceeded Percentage* $P_E$ (right) of different fine-tuned models under varying training sample sizes. *Accuracy* and *Length Exceeded Percentage* are the average values obtained on four datasets.

31.9%. This gap narrows as the training data increases to 80k. Furthermore, incorporating Shift-FFN consistently enhances the performance of the original LoRA model across all data sizes. Even with 80k training samples, the LoRA+Shift-FFN model achieves an average accuracy 0.6% higher than the full fine-tuned model and demonstrates superior performance on all datasets. This experiment further illustrates the scalability of Shift-FFN.

## 4.6 Ablation Studies

Table 3 presents the results of ablation studies, where we evaluate four configurations: (1) w/o $x_{i-1}$, which removes the preceding token's participation in the Editor module, $f_s = W_c\left[ReLU(W_b\,x_i) \odot (W_a\,x_i)\right]$; (2) w/o $x_i$ in gate, which only use the $x_{i-1}$ in the gating mechanism, $f_s = W_c\left[ReLU(W_b\,x_{i-1}) \odot (W_a\,x_{i-1})\right]$; (3) w/o gate, which disables the gating mechanism, $f_s = W_c\,(W_a\,x_{i-1})$; (4) w/o MLP, which directly performs a linear com-

Table 3: Ablation Studies on Qwen2.5-7B-Instruct.

|  | Accuracy ($\uparrow$) | Exceed ($\downarrow$) |
|---|---|---|
| LoRA | 50.4 | 15.0 |
| LoRA+Shift-FFN | **51.2** | **12.7** |
| - w/o $x_{i-1}$ | 50.2 | 14.2 |
| - w/o $x_i$ in gate | 49.8 | 13.8 |
| - w/o gate | 49.3 | 14.3 |
| - w/o MLP | 50.3 | 17.0 |

bination of adjacent tokens, $f_s = tanh(\boldsymbol{w}^T x_{i-1})x_{i-1}$. The experimental results demonstrate that excluding the preceding token leads to performance nearly identical to standard LoRA, indicating that traditional representation learning offers negligible improvement under the LoRA. Furthermore, we find that the gate mechanism that considering both $x_{i-1}$ and $x_i$ is crucial in the Editor module. Without it, performance is even lower than standard LoRA. Thus, dynamically editing representations based on adjacent tokens is vital. It can also be found that performing a linear combination of adjacent tokens without applying MLP to the preceding token doesn't bring any benefit.

## 5 Conclusion

This work finds that fine-tuning LLMs with full parameters or LoRA with a low rank on long CoT data often leads to *Cyclical Reasoning*, where models repeatedly reiterate previous inference steps until the maximum length limit. Investigating the models' internal states, this paper finds that *Cyclical Reasoning* is more likely when the representation differences between adjacent tokens are small. To address this, we propose Shift-FFN, an architecture that introduces an Editor module before the FFN. This module uses the preceding token's representation to edit the current token's representation, thereby dynamically amplifying the representation differences between adjacent tokens. Experimental results demonstrate that LoRA combined with Shift-FFN achieves higher accuracy and a lower rate of *Cyclical Reasoning* across various data sizes compared to full fine-tuning and standard LoRA.

## 6 ETHICS STATEMENT

Our approach does not introduce ethical concerns. The datasets and models we used are public, and there are no privacy issues.

## 7 REPRODUCIBILITY STATEMENT

In this work, we use open-source LLMs and publicly available datasets to conduct our experiments. To ensure reproducibility, we provide the implementation details in the Section 3 and the full code in `https://anonymous.4open.science/r/Shift-FFN`

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

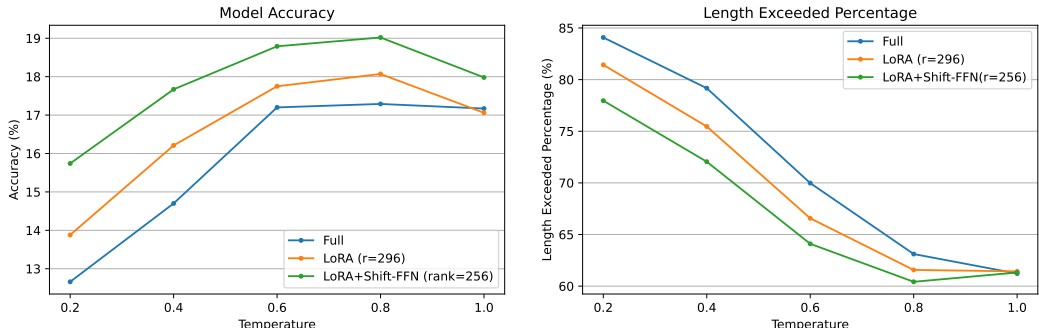

Figure 8: The *Accuracy* (left) and the *Length Exceeded Percentage* (right) of different fine-tuned models for under varying sampling temperatures on AIME24.

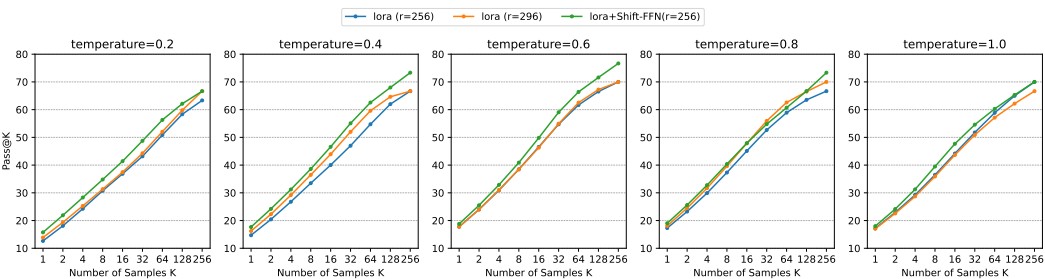

Figure 9: The Pass@K of different fine-tuned models for under varying sampling temperatures on AIME24.

## A  FEEDFORWARD NETWORK

A Transformer language model Vaswani et al. (2017) consists of layers of multi-head self-attention (MHSA) and position-wise feedforward networks (FFN). Each feedforward layer operates independently on individual position vectors in the sequence. The standard FFN can be expressed as follows (bias terms are omitted):

$$\text{FFN}(\boldsymbol{x}_i) = W_{down}[\sigma(W_{up}\,\boldsymbol{x}_i)] \tag{11}$$

where $W_{down} \in \mathbb{R}^{d_m \times d}$ and $W_{up} \in \mathbb{R}^{d \times d_m}$ are parameter matrices, $\boldsymbol{x}_i \in \mathbb{R}^d$ is the representation of token $i$ after MHSA and $\sigma$ represents a nonlinear activation function.

An alternative to the standard FFN is the Gated Linear Unit Shazeer (2020) variant, which has shown improved performance in some scenarios. The GLU-FFN is defined as (bias terms are omitted):

$$\text{FFN}_{\text{GLU}}(\boldsymbol{x}_i) = W_{down}(\sigma(W_{gate}\,\boldsymbol{x}_i) \odot (W_{up}\,\boldsymbol{x}_i)) \tag{12}$$

where $\odot$ denotes element-wise multiplication, and $W_{gate}, W_{up} \in \mathbb{R}^{d \times d_m}$, $W_{down} \in \mathbb{R}^{d_m \times d}$ are parameter matrices. This gating mechanism allows for more flexible information flow and has better performance Shazeer (2020). Contemporary models such as LLaMA Grattafiori et al. (2024) and Qwen Yang et al. (2024) predominantly employ GLU-FFN. Our Shift-FFN can be applied to any type of FFN.

## B  PERFORMANCE UNDER VARYING SAMPLING TEMPERATURES

We also further investigate the impact of sampling temperature on model performance and the rate of *Cyclical Reasoning*. Specifically, we examine the performance of Qwen2.5-7B-Instruct, fine-tuned with different strategies, at sampling temperatures of 0.2, 0.4, 0.6, 0.8, and 1.0. The maximum generation length is set to 16k for computational efficiency. The experimental results are shown in Figure 8. We observe that at lower sampling temperatures, the models exhibit not only lower

accuracy but also a higher *Length Exceeded Percentage*, indicating a greater tendency for *Cyclical Reasoning*. The overall performance of the models appears optimal within the sampling temperature range of 0.6 to 0.8; further increases beyond this range tend to result in a decline in performance. Notably, LoRA+Shift-FFN (r=256) consistently achieves the highest accuracy across all tested sampling temperatures. Based on Figure 9, we also investigate the Pass@K performance of the models at different sampling temperatures. The trend in Pass@K values aligns with the average accuracy observed earlier, with peak performance generally occurring at a temperature of 0.6. In the temperature range of 0.2 to 0.6, LoRA+Shift-FFN (r=256) exhibits a clear advantage in Pass@K. However, this advantage diminishes at higher sampling temperatures (0.8 and 1.0). A potential explanation for this reduction is that the difference in *Length Exceeded Percentage* between the models decreases at these higher temperatures, leading to similar exploration efficiency. Nevertheless, at the optimal sampling temperature of 0.6, LoRA+Shift-FFN (r=256) still demonstrates a significant improvement in Pass@K, validating the effectiveness of our proposed method.

## C  *Cyclical Reasoning* EXAMPLES

Table 4 and 5 show examples of *Cyclical Reasoning* in full fine-tuned Qwen2.5-7B-Instruct on AIME24 and MATH500, alongside correct answers generated by the model with Shift-FFN. It can be found that these problems can actually be solved after generating around 10k tokens, but if the model falls into *Cyclical Reasoning*, it will keep generating tokens until it reaches the 32k length limit, which not only reduces the average accuracy but also seriously affects the reasoning efficiency.

Table 4: Generation comparison of Qwen2.5-7B-Instruct under full and LoRA fine-tuning, where the full fine-tuned model shows *Cyclical Reasoning*.

| **Instruction (AIME24)** | Find the largest possible real part of $(75+117i)z + \frac{96+144i}{z}$ where $z$ is a complex number with $|z| = 4$. |
|---|---|
| Full | <\|begin_of_thought\|>\n\nOkay, so I need to find the largest possible real part of the expression \[(75+117i)z + \frac96+144iz\] where \( z \) is a complex number with \( \|z\| = 4 \). Hmm, let's start by understanding what the problem is asking. 

 ... 
 Therefore, the correct answer is 540.\n\nBut according to the Lagrange multiplier method, it's 518.4. Therefore, there's a contradiction.\n\nThis suggests that the Cauchy-Schwarz approach is incorrect. 

 ... 
 Therefore, the correct answer is 540.\n\nBut according to the Lagrange multiplier method, it's 518.4. Therefore, there's a contradiction.\n\nThis suggests that the Cauchy-Schwarz approach is incorrect. 

 ... 
 Therefore, the correct answer is 540.\n\nBut according 

 Generation Length: 32768 
 Result: Wrong (Reaching the maximum length limit) |
| LoRA+Shift-FFN (rank=256) | <\|begin_of_thought\|>l\n\nOkay, so I need to find the largest possible real part of the expression (75 + 117i)z + (96 + 144i)/z, where z is a complex number with \|z\| = 4. Hmm, let's see. First, I remember that when dealing with complex numbers, 

 ... 
 Expanding and combining like terms, this paper finds the real part of the expression:\n\n\[\n324\cos\theta - 432\sin\theta\n\]\n\nThis real part can be written in the form \(C\cos(\theta + \phi)\), where \(C = \sqrt{324\hat2 + (-432)\hat2} = 540\). The maximum value of this expression is therefore \(540\).\n\nTo verify, we check specific values of \(\theta\) that align the coefficients with the vector \((324, -432)\). This confirms that the maximum real part is indeed \(540\).\n\nThus, the largest possible real part is \(\boxed{540}\).\n\n<end_of_solution> 

 Generation Length: 9902 
 Result: Correct |

Table 5: Generation comparison of Qwen2.5-7B-Instruct under full and LoRA fine-tuning, where the full fine-tuned model shows *Cyclical Reasoning*.

| **Instruction (MATH500)** | The expression $2 \cdot 3 \cdot 4 \cdot 5 + 1$ is equal to 121, since multiplication is carried out before addition. However, we can obtain values other than 121 for this expression if we are allowed to change it by inserting parentheses. For example, we can obtain 144 by writing $(2 \cdot (3 \cdot 4)) \cdot (5 + 1) = 144$. In total, how many values can be obtained from the expression $2 \cdot 3 \cdot 4 \cdot 5 + 1$ by inserting parentheses? (Note that rearranging terms is not allowed, only inserting parentheses). |
|---|---|
| Full | <\|begin_of_thought\|>\n\nOkay, so I need to figure out how many different values can be obtained from the expression 2·3·4·5 + 1 by inserting parentheses. The key here is that changing the grouping with parentheses can alter the order of operations, which might lead to different results. Let me start by recalling the original expression and how parentheses can affect the calculation. ... Wait, no, they are the same. Wait, 2·(3·4·5) = 2·60 = 120, same as (2·3)·4·5 = 6·20 = 120. So different groupings can lead to the same product.\n\nBut inserting parentheses around different parts can lead to different products. For example, 2·(3·4·5) = 120, but (2·3)·(4·5) = 120, but 2·(3·4·5) is different. Wait, no, they are the same. Wait, 2·(3·4·5) = 2·60 = 120, same as (2·3)·4·5 = 6·20 = 120. So different groupings can lead to the same product.\n\nBut inserting parentheses around different parts can lead to different products. For example, 2·(3·4·5) = 120, but (2·3)·(4·5) = 120, but 2·(3·4·5) is different. Wait, no, they are the same. Wait, 2

Generation Length: 32768
Result: Wrong (Reaching the maximum length limit) |
| LoRA+Shift-FFN (rank=256) | <\|begin_of_thought\|>\n\nOkay, so I need to figure out how many different values can be obtained from the expression 2×3×4×5 +1 by inserting parentheses. The problem says that rearranging terms isn't allowed, so we have to keep the original order but group the operations differently with parentheses. ... *7. (2·3·4·5) +1**: \n \[\n 120 + 1 = 121.\n \]\n\n - **8. (2·(3·(4·5))) +1**: \n \[\n 2 \cdot (120) + 1 = 241.\n \]\n\n - **9. (2·3·4) + (5·1)**: \n \[\n 24 + 5 = 29.\n \]\n\n - **10. (2·3)·(4·(5+1))**: \n \[\n 6 \cdot (4 \cdot 6) = 6 \cdot 24 = 144.\n \]\n\n - **11. (2·(3·4))·(5+1)**: \n \[\n 24 \cdot 6 = 144.\n \]\n\n - **12. 2·((3·4)·(5+1))**: \n \[\n 2 \cdot (12 \cdot 6) = 2 \cdot 72 = 144.\n \]\n\n3. **Distinct Values**: \n The distinct results from the valid parenthesizations are: \n - \(121\), \n - \(241\), \n - \(29\), \n - \(144\). \n\nThus, the total number of distinct values obtainable is \(\boxed{4}\).\n\n<\|end_of_solution\|>

Generation Length: 11053
Result: Correct |

