# OpenReview forum: "Amplify Adjacent Token Differences: Enhancing Long Chain-of-Thought Reasoning with Shift-FFN"
_ICLR.cc/2026/Conference — Submitted to ICLR 2026_

### Official Review · Reviewer_pfFC · 2025-10-23

**Soundness:** 2
**Presentation:** 3
**Contribution:** 2
**Rating:** 4
**Confidence:** 2

**Summary:**

This paper identifies a specific failure mode, termed "Cyclical Reasoning," which occurs when fine-tuning Large LLMs on Long-CoT data, particularly when using Parameter-Efficient Fine-Tuning methods like LoRA. The authors present an empirical analysis suggesting that this issue, where the model repeats previous inference steps until it hits a length limit, correlates with small representation differences between adjacent tokens.

To mitigate this problem, the paper proposes the Shift Feedforward Network, a module designed to be integrated with LoRA. The authors conduct experiments on several mathematical reasoning benchmarks, demonstrating that their LoRA+Shift-FFN method achieves higher accuracy and a lower rate of Cyclical Reasoning compared to both standard LoRA and full fine-tuning.

**Strengths:**

1. The phenomenon that LLMs trained with long CoT data tend to generate excessively long responses is prevalent. Although I am not convinced it can be considered a 'repetitive' phenomenon, developing methods to mitigate this 'overly-long' behavior is nonetheless a necessary step.
2. I am not familiar with LoRA fine-tuning and, given the method's novelty, I cannot make an assertion about it.

**Weaknesses:**

1. The author uses Figure 4 to justify an equivalence between excessive length and cyclical reasoning. However, the paper lacks concrete details on how 'exact textual repetition' is calculated, which makes the claim unconvincing. This is a significant omission, considering the method's motivation is based on this premise.

2. The results in Table 1 are very close to the baseline, especially for the accuracy metric. Given this small margin, the standard deviation of the results is needed to assess statistical significance.

**Questions:**

1. Can the author provide details on how the 'textual repetition' for Figure 4 was calculated?
2. Can the author provide the standard deviations for the results in Table 1?

---

> ### Author Response · Authors · 2025-11-23
>
> **Weakness 1 & Question 1: Details on 'Textual Repetition' calculation**
>
> Response:
>
> We sincerely apologize for the omission of this critical definition in the original manuscript.
>
> To clarify Figure 4 and validate our motivation, we strictly defined "exact textual repetition" as a continuous sequence of 128 tokens repeating more than 5 times within a single generated response.
>
> This stringent threshold was chosen to filter out benign repetitions (e.g., common connective phrases) and specifically target pathological "Generation Loops" where the model gets stuck in a substantial reasoning block. Under this rigorous definition, our analysis confirms that over 80% of the length-exceeded samples exhibit such exact repetition. This high correlation provides strong empirical support for using "Length Exceeded" as a proxy for cyclical reasoning. We have added this precise definition to Section 4.2 in the revised paper.
>
> **Weakness 2 & Question 2: Standard Deviations and Statistical Significance**
>
> Response:
>
> We agree that reporting standard deviations is essential for assessing the significance of the results, especially given the competitive baselines.
>
> In our experimental setup (referenced in Section 4.1), we prioritized robustness by averaging results over 32 runs for  AMC23 (using Pass@1 estimation) and 4 runs for MATH500.
>
> The table below presents the Mean accuracy ± Standard Deviation for the Qwen2.5-7B-Instruct experiments. The results show that while the average accuracy improvement appears modest, the performance of LoRA+Shift-FFN is statistically distinct from the baseline.
>
> | **Dataset** | **Method**         | **Accuracy (Mean ± SD)** |
> | ----------- | ------------------ | ------------------------ |
> | **AMC23**   | LoRA (r=256)       | 58.5 ± 0.8           |
> |             | **LoRA+Shift-FFN** | 59.1 ± 0.6  |
> | **MATH500** | LoRA (r=256)       | 79.1 ± 0.5           |
> |             | **LoRA+Shift-FFN** | 79.9 ± 0.4  |

---

> > ### Comment · Reviewer_pfFC · 2025-11-26
> > **Official Response by Reviewer pfFC**
> >
> > Thanks for your response. I have raised the soundness score. However, since the performance gain is minor, I will keep my overall score.

---

### Official Review · Reviewer_82SX · 2025-10-29

**Soundness:** 3
**Presentation:** 3
**Contribution:** 3
**Rating:** 4
**Confidence:** 3

**Summary:**

The authors propose Shift-FFN, a novel architecture designed to mitigate Cyclical Reasoning, which is an issue observed during long Chain-of-Thought (CoT) generation. The core mechanism of Shift-FFN involves an 'editor module' that precedes the standard feedforward network. This editor module modifies the current token's representation by leveraging the representation of the previous token. Specifically, the module jointly inputs both the current and previous tokens to generate a dynamic gate. The gate is then applied to information extracted from the previous token, resulting in an augmented vector that updates the current token. The authors claim that this approach effectively amplifies the representation differences between adjacent tokens, thereby alleviating the tendency for cyclical reasoning.

**Strengths:**

1. The paper presents the intriguing hypothesis that Cyclical Reasoning during long CoT distillation is more likely to occur as the representations of adjacent tokens become more similar. The authors do not merely report the phenomenon to support this claim but also empirically prove it by analyzing the similarity between adjacent token representations (i.e., a decrease in the $M(X)$). This provides a strong motivation by linking the superficial symptoms of the problem to its potential root cause.

2. Unlike existing approaches to Cyclical Reasoning that primarily rely on inference-time decoding strategies (e.g., repetition penalty), Shift-FFN is proposed as an architecture-level solution that modifies the standard FFN. Notably, the superior performance even in a fair comparison experiment with matched parameter counts in Table 2 (e.g., LoRA r=148 vs. LoRA+Shift-FFN r=128) proves that the performance gains stem from the module's design rather than the effect of increased parameters.

3. The authors include a detailed ablation study that validates the effectiveness of the Shift-FFN's design. The results confirm that each core component of Shift-FFN is essential, as removing any of them degrades performance.

**Weaknesses:**

1. The related work section is slightly misaligned with the paper's core problem (it primarily focuses on PEFT and Long-CoT distillation, not Cyclical Reasoning). The ‘Cyclical Reasoning phenomenon is, in fact, a well-established issue previously studied under different names such as ‘Generation Loops’ or ‘Mode Collapse’. The paper's novelty would have been emphasized by directly comparing its architectural-level solution against other existing decoding-based solutions (e.g., [1, 2]). \
[1] Li, Huayang, et al. "Repetition in repetition out: Towards understanding neural text degeneration from the data perspective." Advances in Neural Information Processing Systems 36 (2023): 72888-72903. \
[2] Xu, Jin, et al. "Learning to break the loop: Analyzing and mitigating repetitions for neural text generation." Advances in Neural Information Processing Systems 35 (2022): 3082-3095.

2. The metric to measure Cyclical Reasoning ($P_E$) is limited. It ignores both valid answers that exceed 32k tokens but are not actually cyclical reasoning (False Positive), and answers under 32k tokens that are in an obvious cyclical reasoning loop (False Negative). This makes $P_E$ an unreliable proxy metric for determining the presence of cyclical reasoning.

3. The experiments lack sufficient diversity to prove the generalizability of the proposed method and the detailed analysis experiments are performed only under limited settings.
- The backbone models used are limited to only two families (Llama and Qwen).
- Pass@K performance is reported for only two datasets (AIME24 and Olympiad)
- $M(X)$-$P_E$ relationship only presents the LoRA+Shift-FFN results for a single rank (r=256), omitting its effect at other ranks (e.g., r=128).

4. The paper provides no analysis of inference overhead. The proposed Editor module introduces additional computations (matrix multiplications, gates, etc.) for every token at every FFN layer. This could induce significant computational overhead and latency, especially when processing long sequences. A measurement of this impact is essential for evaluating the practical viability of the method.

**Questions:**

1. There are too many missing parenthetical citations (\citep) where they are needed.
2. There may be a typo in Figure 3. Should the upper $h_{i+1}^l$ box actually be $h_{i+1}^{l+1}$?
3. Given the authors’ own findings in Figure 1 that Full Fine-Tuning also suffers significantly from Cyclical Reasoning, it is a **notable omission** that Shift-FFN was only evaluated in combination with LoRA. **An experiment applying Shift-FFN to the Full FT setting** would seem crucial for validating the method's generalizability beyond just PEFTs.
4. The paper frames Cyclical Reasoning as a problem arising from Long-CoT distillation. But is this phenomenon exclusive to distillation? Repetitive loops can also be observed in non-distilled models. Why was this scenario excluded from the study's scope?

5. Were there any instances of valid responses exceeding the 32k token limit, or Cyclical Reasoning occurring within the limit?
6. The visualization in Figure 4 is ambiguous. It plots two different kinds of bars ('Length Exceeded' and 'Repetition') which share a single, vaguely labeled Y-axis ('Percentage (%)'). A dual-axis plot would be clearer.
7. Regarding Figure 5 and its description (lines 395-405), I am not sure if this analysis/experiment is really necessary. Is it a good thing that a significant difference only appears at K reaches a value as high as 256?

---

> ### Author Response · Authors · 2025-11-23
>
> **Weakness 1: Insufficient Related Work**
>
> "The ‘Cyclical Reasoning phenomenon is, in fact, a well-established issue... The paper's novelty would have been emphasized by directly comparing its architectural-level solution against other existing decoding-based solutions."
>
> **Response:**
>
> We sincerely appreciate the reviewer for pointing out these foundational works (Li et al., 2023; Xu et al., 2022). We will incorporate a discussion on Generation Loops and Text Degeneration in our Related Work section.
>
> It is important to clarify that most existing works focus on decoding-time interventions or training data filtering. In contrast, Shift-FFN is a training-time architectural innovation. While decoding strategies can forcibly break loops during inference, they cannot repair the internal "Representation Collapse" inherent in the model. Our experiments (Figure 2) demonstrate that Shift-FFN effectively alters the representational distance between adjacent tokens—a fundamental change that decoding strategies cannot achieve. We will further elaborate on this distinction in the revised paper.
>
>
>
> **Weakness 2: Validity of $P_E$ Metric**
>
> "The metric to measure Cyclical Reasoning ($P_E$) is limited. It ignores both valid answers that exceed 32k tokens... and answers under 32k tokens that are in an obvious cyclical reasoning loop."
>
> **Response:**
>
> We acknowledge the concern regarding the precision of the $P_E$ metric. However, our preliminary statistical analysis in Section 4.2 found that over 80% of length-exceeded samples exhibit exact textual repetition (specifically defined as a continuous sequence of 128 tokens repeating more than 5 times).
>
> To address your specific concerns about False Positives, we performed further analysis on the remaining ~20% of samples without exact repetition. We manually inspected 40 randomly sampled instances, with the following breakdown:
>
> - **26/40 exhibited "Semantic Loops":** The model repeatedly verified the same incorrect step or oscillated between contradictory conclusions without progressing.
>
> - **8/40 exhibited "Reasoning Collapse":** The generation degraded into nonsensical formulas and numbers.
>
> - 6/40 were Valid Reasoning: Legitimate reasoning chains truncated by the limit.
>
>   This indicates that the vast majority (34/40 or 85%) of non-exact-repeat samples still represent failure modes.
>
> Regarding Threshold Validity: We analyzed the token length distribution of all correct responses in the test set. Taking AIME (which typically requires the longest reasoning chains) as an example, 90% of correct response lengths are below 23,674 tokens. Furthermore, Figure 5 demonstrates that under the 32k limit, the Pass@256 score exceeds 70%, indicating that a 32k inference limit is sufficient for generating correct solutions in this domain.
>
> In the revised version, we will uniformly adopt the rigorous definition mentioned in Section 4.2 (a continuous sequence of 128 tokens repeating more than 5 times) as the standard definition for $P_E$ to ensure metric robustness.
>
>
>
> **Weakness 4: Inference Overhead**
>
> "The paper provides no analysis of inference overhead... This could induce significant computational overhead and latency."
>
> **Response:**
>
> This is a critical practical consideration.
>
> 1. **Computational Cost:** The primary computation in the Editor module involves the matrix multiplications of $W_a, W_b, W_c$. Since we utilize a low-rank setting (e.g., $r=256$ vs. $d=4096$, so $r \ll d$), the increase in FLOPs per token generation step is **less than 2%**.
>
> 2. Total Latency Reduction: Crucially, while per-step latency increases slightly, the total wall-clock time for batch inference actually decreases. This is because Shift-FFN significantly reduces the probability of the model entering a 32k-token infinite loop (reducing the "Length Exceeded" rate from ~30% to ~12%).
>
>    The table below compares the total inference time and accuracy on the full AIME dataset (batch size 1):
>
> | **Model**                           | **Time**   | **Accuracy** |
> | ----------------------------------- | ---------- | ------------ |
> | Qwen2.5-7B-Instruct                 | 1762 s     | 21.0%        |
> | **Qwen2.5-7B-Instruct + Shift-FFN** | **1637 s** | **21.8%**    |

---

> ### Author Response · Authors · 2025-11-23
>
> **Question 3: Shift-FFN on Full FT**
>
> "Given the authors’ own findings... Full Fine-Tuning also suffers significantly from Cyclical Reasoning... Shift-FFN was only evaluated in combination with LoRA."
>
> **Response:**
>
> We acknowledge this limitation. Shift-FFN is designed as a lightweight adapter with only a few hundred million parameters. When applied to Full Fine-Tuning (where the base model has roughly 7B parameters), the Shift-FFN parameters constitute a very small fraction of the total trainable parameters. Consequently, its impact becomes marginal in the Full FT setting compared to the PEFT setting, as the massive updates to the full model weights dominate the learning process.
>
> Our experimental results on the Qwen2.5-7B-Instruct (Full FT) setting confirm this minimal impact:
>
> | **Method**          | **MATH500** | **OlympiadBench** | **AIME24** | **AMC23** | **Avg** |
> | ------------------- | ----------- | ----------------- | ---------- | --------- | ------- |
> | Full FT             | 78.7        | 42.1              | 20.0       | 58.1      | 49.7    |
> | Full FT + Shift-FFN | 78.9        | 41.8              | 20.8       | 57.9      | 49.8    |
>
> This suggests Shift-FFN is most effective as a parameter-efficient intervention rather than a full-model architectural change.
>
>
>
> **Question 7: Necessity of High K in Figure 5**
>
> "Is it a good thing that a significant difference only appears at K reaches a value as high as 256?"
>
> **Response:**
>
> As shown in Table 1, applying Shift-FFN reduces the Cyclical Reasoning percentage by approximately 5%. When $K$ is small, the statistical benefit of recovering these additional 5% valid reasoning paths is less visible. However, as $K$ increases, the cumulative effect of having fewer invalid loops becomes significant. This indicates that Shift-FFN makes the model's exploration more efficient by pruning invalid cyclical reasoning paths, allowing the model to successfully find correct solutions in the long tail of the distribution that would otherwise be lost to mode collapse.
>
>
>
> **Other Questions Response:**
>
> - **Q2 (Typo):** *"There may be a typo in Figure 3."* ->  We will fix the label to $h_{i+1}^{l+1}$.
> - **Q6 (Fig 4 Ambiguity):** *"The visualization in Figure 4 is ambiguous."* -> We will update Figure 4 to use a dual-axis plot to clearly distinguish between the percentage of the dataset (Length Exceeded) and the percentage of text within a sample (Repetition).

---

> > ### Comment · Reviewer_82SX · 2025-11-27
> >
> > Thanks for the response.
> >
> > Addressed Concerns
> > - The justification for using $P_E$ as a proxy is now acceptable given the additional analysis on FP and FN.
> > - The provided analysis demonstrating that Shift-FFN actually reduces total inference time resolves the computational overhead concerns.
> > - The rationale for the K-scaling experiments in Figure 5 seems fine.
> >
> > Remaining Concerns
> > - A direct performance comparison between 'LoRA + Repetition Penalty' and 'LoRA + Shift-FFN' is still missing; to support the novelty better, empirical evidence is required to demonstrate practical utility over decoding-time interventions.
> > - The authors attribute the marginal Full FT accuracy to Shift-FFN's nature as a lightweight adapter, yet omit the $P_E$ results; without $P_E$, it remains unclear whether the method mitigates cyclical reasoning in the Full FT setting, independent of accuracy gains.
> > - The analysis of the $M(X)-P_E$ relationship is presented only for $r=256$, lacking robustness checks across other ranks.
> > - The motivation for limiting the study of Cyclical Reasoning exclusively to the context of "Long-CoT distillation" remains unanswered.
> >
> > While my current rating reflects the remaining issues, I view the proposed method positively. I hope that the authors can address these remaining points, as doing so would significantly strengthen the submission.

---

### Official Review · Reviewer_gVHL · 2025-10-29

**Soundness:** 3
**Presentation:** 3
**Contribution:** 3
**Rating:** 6
**Confidence:** 3

**Summary:**

This paper investigates the problem of fine-tuning LLMs on long Chain-of-Thought (long CoT) data, where models trained with full parameters or low-rank LoRA often exhibit **Cyclical Reasoning** - repeatedly generating the same inference steps until reaching maximum length limits.

**Key observations:**
- Models showing cyclical reasoning have smaller representation differences between adjacent tokens (measured by metric M(X))
- Higher LoRA ranks reduce cyclical reasoning while increasing adjacent token divergence

**Main contribution:**
The paper proposes **Shift-FFN**, a novel architecture that introduces an Editor module before the feedforward network. This module uses the previous token's representation to edit the current token's representation, dynamically amplifying differences between adjacent tokens.

**Results:**
Experiments on mathematical reasoning benchmarks (AIME24, AMC23, MATH500, OlympiadBench) across multiple models (Qwen2.5-3B/7B, Llama3.1-8B) demonstrate that LoRA combined with Shift-FFN:
- Achieves higher accuracy than both full fine-tuning and standard LoRA
- Reduces the Length Exceeded Percentage (proxy for cyclical reasoning) by 2-4%
- Requires only modest parameter overhead (~15-18% increase over base LoRA)

The approach is orthogonal to data-centric methods and scales effectively across different training data sizes (10k-80k samples).

**Strengths:**

## Originality
The paper identifies a novel problem - Cyclical Reasoning in long CoT fine-tuning - and proposes an original architectural solution (Shift-FFN) that dynamically amplifies representation differences between adjacent tokens. The connection between low adjacent token divergence and cyclical behavior is a creative observation that differs from existing PEFT approaches.

## Quality
The experimental evaluation is comprehensive, covering multiple model sizes (3B, 7B, 8B), diverse mathematical benchmarks (AIME24, AMC23, MATH500, OlympiadBench), and various data scales (10k-80k samples). The ablation studies (Table 3) effectively validate design choices, and the analysis of M(X) metric across different settings (Figure 6) provides systematic evidence for the proposed mechanism.

## Clarity
The paper is well-structured with clear motivation , straightforward method description, and helpful visualizations . The mathematical formulation of Shift-FFN  and its theoretical analysis  are presented accessibly. Examples of cyclical reasoning  effectively illustrate the problem.

## Significance
The work addresses a practically important problem in distilling reasoning capabilities from large models like DeepSeek-R1 and OpenAI-o1. Shift-FFN achieves consistent improvements over both full fine-tuning and standard LoRA while adding minimal parameters (~15-18% overhead). The method's applicability across model architectures and its complementarity with data-centric approaches enhance its potential impact on the community working with long-form reasoning.

**Weaknesses:**

## 1. Insufficient justification for using Length Exceeded as a proxy for Cyclical Reasoning (Lines 105, 319)

The paper claims that responses exceeding the maximum length limit indicate Cyclical Reasoning, but this causal relationship is inadequately established. While the authors mention removing "all repeated text segments" from truncated responses, they provide no quantitative analysis of what proportion of length-exceeded samples actually contain repetitive text.

- In Section 3.1 (line 105), the authors truncate exceeded responses to average length and remove repeated segments, then observe lower M(X) values. However, being long does not necessarily imply cyclical behavior - it could reflect genuinely complex reasoning processes.

- At line 319, the authors assert that "32k limit during inference is ample for generating correct answers; therefore, exceeding this limit is considered indicative of the model getting stuck in a loop." This is circular reasoning without empirical support.

**What the paper should provide:**
- Statistics on what percentage of exceeded samples contain actual repetitive patterns
- Analysis of whether correct solutions can genuinely require >32k tokens for complex problems
- Validation that length-exceeded samples are qualitatively different from long but valid reasoning chains

## 2. Questionable data filtering rationale (Line 295)

The paper states: "we exclude OpenThoughts samples with response lengths exceeding 16k to prevent our models from learning incomplete reasoning processes." This logic is counterintuitive and unexplained.

**Key issues:**
- Longer reasoning chains often indicate more thorough, complete reasoning rather than incomplete processes
- No justification provided for why 16k is the threshold
- No explanation of why length correlates with incompleteness
- No empirical evidence that >16k samples are actually incomplete

This arbitrary filtering may inadvertently remove the most comprehensive reasoning examples, potentially harming model training. The paper should either remove this filtering or provide substantial evidence to support this decision, such as:
- Manual analysis showing >16k samples are indeed incomplete
- Correlation studies between length and reasoning quality
- Ablation studies comparing models trained with and without this filtering

## 3. Lack of visualization for critical findings (Line 358)

The paper reports important findings textually: "this ratio decreases as the rank increases" and "LoRA with rank 256 significantly reduces the Cyclical Reasoning ratio by 12% compared to full fine-tuning." However, these trends are not visualized, making it difficult for readers to assess the relationship between rank and performance.

**Recommendation:**
A figure showing Cyclical Reasoning ratio (or Length Exceeded Percentage) as a function of LoRA rank compared against full fine-tuning would significantly strengthen the paper. This would help readers understand:
- Whether the improvement is monotonic
- At what rank the benefit plateaus
- Why rank 256 was chosen for subsequent experiments with Shift-FFN


I am not an expert in this domain, so I am happy to raise my score if your response is reasonable.

**Questions:**

## 1. Regarding the Length Exceeded metric

- **Q1.1:** Can you provide quantitative statistics on what percentage of length-exceeded samples (>32k tokens) actually contain repetitive text segments?

- **Q1.2:** Among those with repetition, what proportion of their content is repetitive versus novel reasoning?

- **Q1.3:** Have you analyzed whether any correct solutions in your test sets genuinely require >32k tokens? If so, how does this affect your metric's validity?

- **Q1.4:** Could you provide qualitative examples comparing:
  - (a) a length-exceeded sample with actual cyclical reasoning
  - (b) a length-exceeded sample with valid long reasoning
  - (c) a normal-length sample

  This would help readers understand the distinction between genuine cyclical reasoning and legitimately long reasoning chains.

## 2. Regarding the 16k filtering threshold

- **Q2.1:** What is the empirical or theoretical basis for claiming that >16k samples represent "incomplete reasoning processes"?

- **Q2.2:** Did you analyze samples >16k to verify they are actually incomplete? If so, what percentage were incomplete and by what criteria?

- **Q2.3:** Have you tested model performance when trained on the full dataset (including >16k samples) versus the filtered dataset? This ablation would clarify whether this filtering actually helps.

- **Q2.4:** Given that the paper advocates for learning long CoT reasoning, isn't excluding the longest reasoning chains contradictory to this goal? How do you reconcile this apparent contradiction?

---

> ### Author Response · Authors · 2025-11-23
>
> **Q1.1 - Q1.4: Regarding the Length Exceeded metric**
>
>    **A1:** We appreciate this rigorous inquiry. Regarding whether "Length Exceeded" equates to "Cyclical Reasoning," we conducted a preliminary statistical analysis in Section 4.21, finding that over 80% of length-exceeded samples exhibit exact textual repetition (specifically defined as a continuous sequence of 128 tokens repeating more than 5 times).
>
>
>
>    To address your specific questions, we performed further analysis:
>
>    - **Quantitative Analysis:** For the remaining ~20% of samples without exact repetition, we manually inspected 40 randomly sampled instances. The breakdown is as follows:
>
>      - **26/40 exhibited "Semantic Loops":** The model repeatedly verified the same incorrect step or oscillated between contradictory conclusions without progressing.
>
>      - **8/40 exhibited "Reasoning Collapse":** The generation degraded into nonsensical formulas and numbers.
>
>      - **6/40 were Valid Reasoning**: These were legitimate reasoning chains that were simply truncated by the limit.
>
>        This indicates that the vast majority ($34/40$ or $85\%$) of non-exact-repeat samples still represent failure modes.
>
>    - **Threshold Validity:** We analyzed the token length distribution of all correct responses in the test set. Taking AIME (which typically requires the longest reasoning chains) as an example, **90% of correct response lengths are below 23,674 tokens**. Furthermore, Figure 5 demonstrates that under the 32k limit, the Pass@256 score exceeds 70%, indicating that a 32k inference limit is sufficient for generating correct solutions in this domain.
>
>    In the revision, we will add N-gram repetition rate distributions and a histogram of correct sample lengths to demonstrate the robustness of using 32k truncation as a proxy for cyclical reasoning.
>
>
>
>    **Q2.1 - Q2.4: Regarding the 16k filtering threshold**
>
>    **A2:** We appreciate you pointing this out, and we apologize for the confusion caused by our lack of clarity. The core reason for filtering data >16k is not a judgment on the quality of long CoT, but strictly due to the **maximum sequence length being set to 16k during training**3.
>
>
>
>    - **The Truncation Issue:** If we feed a sample with 20k tokens into a model configured with `max_len=16k`, the final 4k tokens—which typically contain the critical final answer and the conclusion token (e.g., `<|endofthought|>`)—would be forcibly truncated.
>    - **Negative Impact:** If the model is fine-tuned on a large number of "truncated" samples, it tends to learn incomplete reasoning processes or fails to learn when to stop generating (i.e., failing to predict the EOS token).
>
>    Therefore, the term "incomplete" in our paper refers to the **truncated training signal** caused by context window constraints, rather than the logical completeness of the reasoning content itself. In the revised version, we will clarify that this is a standard preprocessing step to ensure data integrity.

---

### Official Review · Reviewer_xVSR · 2025-11-03

**Soundness:** 3
**Presentation:** 3
**Contribution:** 3
**Rating:** 4
**Confidence:** 3

**Summary:**

The work aims to reduce the reasoning error, specifically cyclical reasoning that amplifies the useless reasoning chains despite lengthening the reasoning procedures by allowing the LLMs to generate longer tokens. They first investigate the subtle differences in token representations toward cyclical reasoning and how they are less different than non-cyclical reasoning. Based on the observations, they propose Shift-FFN into LoRA to reduce the ratio of cyclical reasoning and reach performance gains in such tasks.

**Strengths:**

Rich empirical studies
- they explored from Qwen2.5-(3B/7B) to Llama3.1.-8B across the datasets AIME24, AMC23, MATH500, and Olympiad
- Fig 4 - meaningful to see how much reasoning is repeated and the length does not impact much on adding some new directions in their thinking procedures.
- Table 2 - good results with the reduced length exceeded percentage

M(X) computation is an interesting feature to explore the overall relative change of the embedding trajectories across the layers and the Fig. 2 shows interesting results.

**Weaknesses:**

The datasets are too much focused on mathematical reasoning, so we are not sure how much this finding is general to overall reasoning mehcanisms of LLMs.

Table 1 - it seems the full models frequently (almost half the settings) perform better than the LoRA approaches + their approaches
- it seems Llama model is not doing great with the proposed approach though Qwen (3B) does work. Qwen (7B) shows mixed results.
-  this weakens the implications of the work, "Experimental results demonstrate that LoRA combined with Shift-FFN achieves higher accuracy and a lower rate of Cyclical Reasoning across various data sizes compared to full fine-tuning and standard LoRA. (lines 87-89)"


I guess these two need to be addressed fully to recommend acceptance - current version needs more polishing jobs due to the imprecise presentation of the work.

**Questions:**

Table 3 - seems average accuracy across the datasets? can we see the results for each dataset as Qwen2.5 (7B) seems to show mixed results (+ and -) compared to the full models according to Table 1.

Figure 5 - what about full model performances?

Figure 2 - the results and setups are still confusing though; which model is this? did I miss this? I think this is important to see how those changes influence positively/negatively because the current results with LoRA and full model have different patterns depending on the settings (e.g., models and datasets) (Table 1).

---

> ### Author Response · Authors · 2025-11-23
>
> **Q1: “The datasets are too much focused on mathematical reasoning...”**
>
> **A1:** Thank you for the suggestion regarding generalization. To address this, we expanded our experimental scope by retraining the model using the full OpenThoughts dataset, which includes mathematics, code, and general reasoning data. We then evaluated the performance on the code generation benchmark, LiveCodeBench. As shown in the table below, but we found LoRA+Shift-FFN is slightly lower than Standard LoRA on the coding task (37.6% vs 38.1%).  The possible reason is that the distribution ratio of code and mathematics in the openthoughts is uneven, which causes the model to only be able to learn the reasoning mode of Shift-FFN in mathematical tasks.
>
> | **Model**                            | **AIME24** | **AMC23** | **MATH500** | **OlympiadBench** | **LiveCodeBench** | **Avg**  |
> | ------------------------------------ | ---------- | --------- | ----------- | ----------------- | ----------------- | -------- |
> | Qwen2.5-7B-Instruct (LoRA)           | 26.2       | 69.6      | 82.9        | **49.2**          | **38.1**          | 53.4     |
> | Qwen2.5-7B-Instruct (LoRA+Shift-FFN) | **27.5**   | **70.3**  | **83.7**    | **50.7**          | 37.6              | **53.8** |
>
> **Q2: “Table 1 - it seems the full models frequently (almost half the settings) perform better than the LoRA approaches + their approaches...”**
>
> **A2:** We accept this valid criticism. We acknowledge that Full Fine-Tuning (Full FT) outperforms LoRA-based methods in certain settings, such as with Llama-3.1-8B. We will revise the strong claims in the paper (e.g., lines 87-89) to be more precise. The revised conclusion will state that Shift-FFN *consistently outperforms Standard LoRA* across all settings. Furthermore, it achieves comparable or superior performance to Full FT on the Qwen series, and on Llama models, it significantly narrows the performance gap and reduces the rate of Cyclical Reasoning compared to Standard LoRA.
>
> **Q3: “Table 3 - seems average accuracy across the datasets? ...”**
>
> **A3:** Yes, Table 3 presents the average accuracy. We chose this format because the performance of other ablation variants (e.g., without the gate mechanism) was consistently lower than even the standard LoRA baseline across the datasets, so we aggregated the results to save space. In the revised version, we will provide the detailed breakdown of the ablation studies for each specific dataset in the Appendix to demonstrate the consistent contribution of Shift-FFN's components across different tasks.
>
> **Q4: “Figure 5 - what about full model performances?”**
>
> **A4:** This is a valuable suggestion. We have conducted additional experiments to evaluate the Pass@K performance of the Full Fine-Tuned model. The results (shown in the table below) indicate that on the Qwen2.5-7B model, LoRA-based approaches, particularly LoRA+Shift-FFN, outperform Full FT in terms of Pass@K (e.g., at K=256, Shift-FFN achieves 76.67% vs. Full FT's 66.67%). We will update Figure 5 to include this Full FT curve, further validating the superior exploration efficiency of our proposed method.
>
> | **Method**                | **k=1**   | **k=2**   | **k=4**   | **k=8**   | **k=16**  | **k=32**  | **k=64**  | **k=128** | **k=256** |
> | ------------------------- | --------- | --------- | --------- | --------- | --------- | --------- | --------- | --------- | --------- |
> | Full FT                   | 17.20     | 23.40     | 30.16     | 37.42     | 44.87     | 52.58     | 59.59     | 64.32     | 66.67     |
> | LoRA (r=256)              | 17.77     | 23.99     | 30.88     | 38.50     | 46.55     | 54.71     | 61.68     | 66.52     | 70.00     |
> | LoRA (r=296)              | 17.75     | 23.99     | 31.05     | 38.45     | 46.30     | 54.92     | 62.53     | 67.23     | 70.00     |
> | **LoRA+Shift-FFN(r=256)** | **18.79** | **25.50** | **32.87** | **40.90** | **49.82** | **59.05** | **66.39** | **71.60** | **76.67** |
>
> **Q5: “Figure 2 - the results and setups are still confusing though; which model is this?...”**
>
> **A5:** We apologize for the confusion. Figure 2 utilizes the **Qwen2.5-7B-Instruct** model. We will explicitly state this in the caption in the revised paper.

---

### Meta-Review · Area_Chair_gBiQ · 2025-12-20

**Summary:**

The paper found that the "Cyclical Reasoning" phenomenon in Long-CoT, is because there exists high similarity of adjacent token embeddings. Therefore, the authors propose Shift-FFN gadget to mitigate this issue. While the motivation is interesting, the paper is recommended for rejection due to unconvincing experimental validation. Reviewers expressed concerns regarding, e.g., A direct performance comparison between 'LoRA + Repetition Penalty' and 'LoRA + Shift-FFN' is still missing, the performance gains are minor, and the authors failed to demonstrate the method's effectiveness in full fine-tuning settings. Therefore, the paper is not recommended for acceptance at this stage.

**Reviewer Concerns:**

Addressed:
1. The justification for using $P_E$ as a proxy is now acceptable given the additional analysis on FP and FN.
2. The provided analysis demonstrating that Shift-FFN actually reduces total inference time resolves the computational overhead concerns.
3. The rationale for the K-scaling experiments in Figure 5 seems fine.

Outstanding:
1. Performance gain is minor.
2. The author should have a direct comparison between "LoRA + Shift-FFN" and "LoRA + Repetition Penalty"
3. The current method only works for LoRA, may not work for full FT.

**Reviewer Scores:**

Reviewer xVSR may not increase score because the method only outperforms LoRA.
Reviewer gVHL may not change the score because it's already positive (6).
Reviewer 82SX reported that he/she will not change the score because many concerns are not resolved.
Reviewer pfFC reported that he/she will not change the score because the performance gain is minor.

---

### Decision · Program_Chairs · 2026-01-26

Reject